# ROBUSTMIX: IMPROVING ROBUSTNESS BY REGULARIZING THE FREQUENCY BIAS OF DEEP NETS

## ABSTRACT

Deep networks have achieved impressive results on a range of well curated benchmark datasets. Surprisingly, their performance remains sensitive to perturbations that have little effect on human performance. In this work, we propose a novel extension of Mixup called Robustmix that regularizes networks to classify based on lower frequency spatial features. We show that this type of regularization improves robustness on a range of benchmarks such as Imagenet-C and Stylized Imagenet. It adds little computational overhead and furthermore does not require a priori knowledge of a large set of image transformations. We find that this approach further complements recent advances in model architecture and data augmentation attaining a state-of-the-art mean corruption error (mCE) of 44.8 with an EfficientNet-B8 model and RandAugment, which is a reduction of 16 mCE compared to the baseline.

## 1 INTRODUCTION

Deep neural networks have achieved state-of-the-art accuracy across a range of benchmark tasks such as image segmentation (Ren et al., 2015) and speech recognition (Hannun et al., 2014). These successes have led to the widespread adoption of neural networks in many real-life applications. However, while these networks perform well on curated benchmark datasets, their performance can suffer greatly in the presence of small data corruptions (Szegedy et al., 2014; Goodfellow et al., 2014; Moosavi-Dezfooli et al., 2017; Athalye et al., 2018; Hendrycks & Dietterich, 2018). This poses significant challenges to the application of deep networks.

Hendrycks & Dietterich (2018) show that the accuracy of a standard model on Imagenet can drop from 76% to 20% when evaluated on images corrupted with small visual transformations. This shows modern networks are not robust to certain small shifts in the data distribution. That is a concern because such shifts are common in many real-life applications. Secondly, Szegedy et al. (2014) show the existence of *adversarial* perturbations which are imperceptible to humans but have a disproportionate effect on the predictions of a network. This raises significant concerns about the safety of using deep networks in critical applications such as self driving cars (Sitawarin et al., 2018).

These problems have led to numerous proposals to improve the robustness of deep networks. Some of these methods such as those proposed by Hendrycks et al. (2019) require a priori knowledge of the visual transformations in the test domain. Others, such as Geirhos et al. (2018) use a deep network to generate transformations which comes with significant computation cost.

This paper proposes a new data augmentation technique to improve the robustness of deep networks by regularizing frequency bias. This new regularization technique is based on Mixup and has many advantages compared to related robustness regularizers: (1) it does not require knowledge of a large set of priori transformations, (2) it is inexpensive and (3) it doesn't have many hyper-parameters. The key idea is to bias the network to rely more on lower spatial frequencies to make predictions.

We demonstrate on Imagenet-C that this method works well with recent advances and reaches a state-of-the-art mCE of 44.8 with 85.0 clean accuracy with Efficientnet-B8 and RandAugment(Cubuk et al., 2019). This is an improvement of 16 mCE compared to the baseline Efficientnet-B8 and matches ViT-L/16 (Dosovitskiy et al., 2020), which is trained on $300\times$ more data. We find that our implementation of the method with DCT transform adds negligible overhead in our experiments. We

find that Robustmix improves accuracy on Stylized-Imagenet by up to 15 points and we show that it can increase adversarial robustness.

## 2 RELATED WORK

The proposed approach can be seen as a generalization of Mixup (Zhang et al., 2018). Mixup is a data augmentation method that regularizes models to behave more linearly between examples. It does so by training the model on linear interpolations of two input examples and their respective labels. These new examples are generated as follows

$$\tilde{x} = \texttt{mix}(x_1, x_2, \lambda), \qquad \text{where } x_1, x_2 \text{ are input images}$$
$$\tilde{y} = \texttt{mix}(y_1, y_2, \lambda), \qquad \text{where } y_1, y_2 \text{ are labels}$$

with `mix` being the linear interpolation function

$$\texttt{mix}(x_1, x_2, \lambda) = \lambda x_1 + (1 - \lambda)x_2 \tag{1}$$

where $\lambda \sim \text{Beta}(\alpha, \alpha)$, $\alpha$ is the Mixup coefficient hyper-parameter.

Zhang et al. (2018) show that Mixup improves the accuracy of networks and can also improve the robustness of the network.

Augmix (Hendrycks et al., 2019) is a data augmentation technique to improve robustness by training on a mix of known image transformations. This method add little computational overhead, but requires knowledge of a diverse set of domain specific transformations. Hendrycks et al. (2019) mixes a set of 9 different augmentation to reach $68.4$ mCE on Imagenet. In contrast, the proposed method does not rely on specific image augmentations and instead relies on the more general principle that natural images are a kind of signal where most of the energy is concentrated in the lower frequencies.

Zhang (2019) uses low pass filters directly inside the model to improve the frequency response of the network. Our method also makes use of low-pass filtering but does not completely remove high frequency features. Additionally, we only uses frequency filtering during training and therefore no computational overhead is incurred during evaluation.

## 3 METHOD

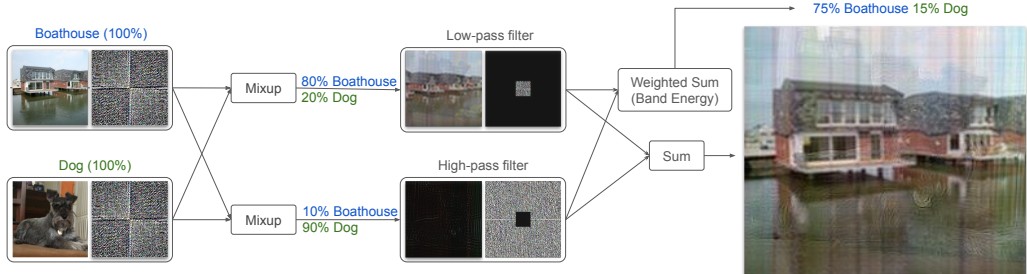

Figure 1: Illustration of the method. In order to better illustrate the method, we display the Fourier spectrum of the images next to them. We can see that even though 90% of the higher frequencies belong to the image of a dog, Robustmix assigns more weight to the boathouse label because it assigns more weight to the lower frequencies.

In this section, we introduce a novel extension of Mixup called *Robustmix* that increases robustness by regularizing the network to focus more on the low frequency features in the signal.

**Motivation** Wang et al. (2020) suggest that convolutional networks trade robustness for accuracy in their use of high frequency image features. Such features can be perturbed in ways that change the prediction of the model even though humans cannot perceive the change. This can lead models to

make puzzling mistakes such as with adversarial examples. Our aim is to increase robustness while retaining accuracy by regularizing how high frequency information is used by the model.

**Robustmix** We propose to regularize the sensitivity of the model to each frequency band by extending Mixup's linear interpolations with a new type of band interpolation. The key insight is that we can condition the sensitivity to each band using images that mix the frequency bands of two different images. Suppose that we mix the lower frequency band of an image of a boathouse with the high frequency band of an image of a dog. We can encourage sensitivity to the lower band by training the model to predict dog for this mixed image. However, this approach is too simplistic because it completely disregards the impact of the image in the high band.

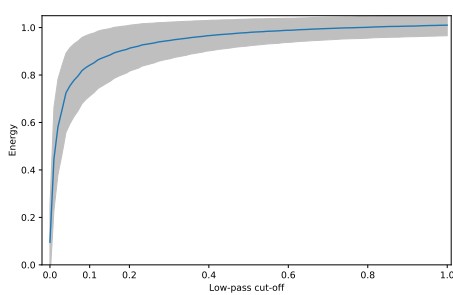

Instead, we interpolate the label of such mixed images according to an estimate of the importance of each frequency band. We propose to use the relative amount of energy in each band as an estimate of the importance. Thus the sensitivity of the model to high frequency features will be proportional to their energy contribution in natural images. And as we can see in Figure 2, most of the spectral energy in natural images is concentrated in the lower end of the spectrum. This should limit the ability of high frequency perturbations to unilaterally change the prediction.

Figure 2: Plot of the cumulative energy in Imagenet images as a function of the frequency cutoff.

Furthermore, we use linear interpolations of images like in mixup within each band instead of raw images. This closely reflects the more common case where the features in the bands are merely corrupted instead of entirely swapped. It also has the benefit of encouraging linearity inside the same frequency band.

Specifically, the mixing formula for Robustmix is given by

$$\tilde{x} = \texttt{Low}(\texttt{mix}(x_1, x_2, \lambda_L), c) + \texttt{High}(\texttt{mix}(x_1, x_2, \lambda_H), c)$$
$$\tilde{y} = \lambda_c\texttt{mix}(y_1, y_2, \lambda_L) + (1 - \lambda_c)\texttt{mix}(y_1, y_2, \lambda_H)$$

where $\lambda_L, \lambda_H \sim \text{Beta}(\alpha, \alpha)$, $\alpha$ is the Mixup coefficient hyper-parameter, and $\texttt{Low}(\cdot, c), \texttt{High}(\cdot, c)$ are a low pass and high pass filter respectively with a uniformly sampled cutoff frequency $c \in [0, 1]$. And $\lambda_c$ is the coefficient that determines how much weight is given to the lower frequency band. It is given by the relative amount of energy in the lower frequency band for natural images

$$\lambda_c = \frac{E[\|\texttt{Low}(x_i, c)\|^2]}{E[\|x_i\|^2]}. \tag{2}$$

This coefficient can be efficiently computed on a mini-batch of examples.

**Implementation** Computational overhead is an important consideration for data augmentation techniques since training deep networks is computationally intensive and practitioners have limited computational budget. We note that many popular techniques such as Mixup (Zhang et al., 2018) add little overhead.

The frequency separation is implemented using a Discrete Cosine Transform (DCT) to avoid the complex multiplication required by an Discrete Fourier Transform. We multiply the images with the 224x224 DCT matrix directly because the spatial dimensions are relatively small and (non-complex) matrix multiplication is well-optimized on modern accelerators. A batch of images is transformed into frequency space and the low and high pass filtered images must be transformed back to image space. Additionally, we must apply the DCT transform over the x and y dimension separately. Thus, 6 DCT matrix multiplications are required which results in $0.2$ GFLOPs per image. In contrast, just the forward pass of ResNet50 requires $3.87$ GFLOPs (Hasanpour et al., 2016).

In our implementation of Robustmix, we reorder commutative operations (low pass and mixing) in order to compute the DCT only a single time per minibatch. The pseudocode is provided in Algorithm 1, where reverse is a function that reverses the rows of its input matrix.

---

**Algorithm 1** Robustmix

---

**Input:** Minibatch of inputs $X \in \mathbb{R}^{N \times H \times W \times D}$ and labels $Y \in \mathbb{R}^{N \times C}$, $\alpha \in \mathbb{R}$
**Output:** Augmented minibatch of inputs $\tilde{X} \in \mathbb{R}^{N \times W \times H \times D}$ and labels $\tilde{Y} \in \mathbb{R}^{N \times C}$

$\quad \lambda_L, \lambda_H \sim \text{Beta}(\alpha, \alpha)$ and $c \sim U(0, 1)$
$\quad L \leftarrow \text{Low}(X, c)$
$\quad H \leftarrow 1 - L$
$\quad \lambda_c \leftarrow \frac{\|L\|^2}{\|X\|^2}$
$\quad \tilde{X} \leftarrow \text{mix}(L, \text{reverse}(L), \lambda_L) + \text{mix}(H, \text{reverse}(H), \lambda_H)$
$\quad \tilde{Y} \leftarrow \text{mix}(Y, \text{reverse}(Y), \lambda_c * \lambda_L + (1 - \lambda_c) * \lambda_H)$

---

## 4 RESULTS

### 4.1 DATASETS AND METRICS

**ImageNet.** ImageNet (Deng et al., 2009) is a classification dataset that contains 1.28 million training images and 50000 validation images with 1000 classes. We evaluate the common classification accuracy which will be referred to as clean accuracy. We use the standard Resnet preprocessing resulting in images of size 224x224 (He et al., 2015). The standard models, without any additional data augmentation process, will be qualified as the baseline.

**ImageNet-C.** This dataset is made of 15 types of corruption drawn from four main categories: noise, blur, weather and digital (Hendrycks & Dietterich, 2018). These corruptions are applied to the validation images of ImageNet at 5 different intensities or levels of severity. Following (Hendrycks & Dietterich, 2018), we evaluate the robustness of our method by reporting its **mean corruption error (mCE)** normalized with respect to AlexNet errors:

$$\text{mCE} = \frac{\sum\limits_{\text{corruption } c} \text{CE}_c}{\text{Total Number of Corruptions}}, \text{ with } \text{CE}_c = \frac{\sum\limits_{\text{severity } s} E_{c,s}}{\sum_s E_{c,s}^{\text{AlexNet}}}$$

**Stylized-ImageNet.** Stylized-ImageNet (SIN) is constructed from ImageNet by replacing the texture in the original image using style transfer, such that the texture gives a misleading cue about the image label (Geirhos et al., 2018). The 1000 classes from ImageNet are reduced to 16 shape categories, for instance all labels for dog species are grouped under one dog label, same for chair, car, etc. There are 1280 generated cue conflict images (80 per category). With SIN, we evaluate the classification accuracy (SIN accuracy) and measure the model's shape bias. Following Geirhos et al. (2018), the model's bias towards shape versus texture is measured as

$$\text{shape bias} = \frac{\text{correct shapes}}{\text{correct shapes + correct textures}}.$$

### 4.2 EXPERIMENTAL SETUP

We chose to do evaluations on residual nets (ResNet-50 and ResNet-152) and EfficientNets (EfficientNet-B0, EfficientNet-B1, EfficientNet-B5 and EfficientNet-B8). Experiments were run on 8x8 TPUv3 instances for the the bigger EfficientNets (EfficientNet-B5 and EfficientNet-B8); and the other experiments were run on 4x4 TPUv3 slices. For the Resnet models, we use the same standard training setup outlined in Goyal et al. (2017). However, we use cosine learning rate Loshchilov & Hutter (2016) with a single cycle for Resnets that are trained for 600 epochs.

### 4.3 ROBUSTNESS RESULTS

**Imagenet-C** First, we evaluate the effectiveness of the proposed method in improving robustness to the visual corruptions considered in Imagenet-C. In Table 1, we can see that Robustmix consistently improves robustness to the considered transformations, with a 15 point decrease in mCE over the baseline for ResNet-50. Robustmix with ResNet-50 achieves 61.2 mCE without degrading accuracy on the clean dataset compared to the baseline. In fact, we find a small improvement over the baseline

of 0.8% on the clean error. While Mixup yields a larger gain of 1.9% on the clean accuracy, we find that Robustmix improves mCE by up to 6 points more than Mixup. These results also compare favorably to Augmix, which needs to be combined with training on Stylized ImageNet (SIN) to reduce the mCE by 12 points. And this improvement comes at significant cost to the accuracy due to the use of the Stylized Imagenet dataset. We also observe a similar trade-off between accuracy and robustness as we can observe in Figure 3. We observe that Mixup consistently produces lower clean error for smaller models, but the accuracy gap with Robustmix disappears as the model gets bigger.

While it is not directly comparable to ViT-L/16 due to its use of $300\times$ more data, we see that Efficientnet-B8 with Robustmix and RandAugment has better robustness at $44.8$ mCE. It is also competitive with DeepAugment (Hendrycks et al., 2020) which requires training additional specialized image-to-image models on tasks such as super-resolution to produce augmented images. By comparison, our approach does not rely on extra data or extra trained models.

| Method | Clean Accuracy | mCE | Size | Extra Data |
|---|---|---|---|---|
| ResNet-50 Baseline (200 epochs) | 76.3 | 76.9 | 26M | 0 |
| ResNet-50 Baseline (600 epochs) | 76.3 | 78.1 | 26M | 0 |
| ResNet-50 BlurPool (Zhang, 2019) | 77.0 | 73.4 | 26M | 0 |
| ResNet-50 Mixup (200 epochs) | 77.5 | 68.1 | 26M | 0 |
| ResNet-50 Mixup (600 epochs) | **78.2** | 67.5 | 26M | 0 |
| ResNet-50 Augmix | 77.6 | 68.4 | 26M | 0 |
| ResNet-50 Augmix + SIN | 74.8 | 64.9 | 26M | 0 |
| ResNet-50 Robustmix (600 epochs) | 77.1 | **61.2** | 26M | 0 |
| EfficientNet-B0 Baseline | 76.8 | 72.4 | 5.3M | 0 |
| EfficientNet-B0 Mixup ($\alpha = 0.2$) | **77.1** | 68.3 | 5.3M | 0 |
| EfficientNet-B0 Robustmix ($\alpha = 0.2$) | 76.8 | **61.9** | 5.3M | 0 |
| EfficientNet-B1 Baseline | 78.1 | 69.4 | 7.8M | 0 |
| EfficientNet-B1 Mixup ($\alpha = 0.2$) | **78.9** | 64.7 | 7.8M | 0 |
| EfficientNet-B1 Robustmix ($\alpha = 0.2$) | 78.7 | **57.8** | 7.8M | 0 |
| EfficientNet-B5 Baseline | 82.7 | 65.6 | 30M | 0 |
| EfficientNet-B5 Mixup ($\alpha = 0.2$) | 83.3 | 58.9 | 30M | 0 |
| EfficientNet-B5 Robustmix ($\alpha = 0.2$) | 83.3 | 51.7 | 30M | 0 |
| EfficientNet-B5 RandAug+Robustmix ($\alpha = 0.2$) | **83.8** | **48.7** | 30M | 0 |
| BiT m-r101x3 (Kolesnikov et al., 2020) | 84.7 | 58.27 | 387.9M | 12.7M |
| ResNeXt-101 $32 \times 8d$+DeepAugment+AugMix (Hendrycks et al., 2020) | 79.9 | **44.5** | 88.8M | Extra models |
| ViT-L/16 (Dosovitskiy et al., 2020) | **85.2** | 45.5 | 304.7M | 300M |
| EfficientNet-B8 Baseline | 83.4 | 60.8 | 87.4M | 0 |
| EfficientNet-B8 Robustmix ($\alpha = 0.4$) | 84.4 | 49.8 | 87.4M | 0 |
| EfficientNet-B8 RandAug+Robustmix ($\alpha = 0.4$) | **85.0** | **44.8** | 87.4M | 0 |

Table 1: Comparison of various models based on Imagenet accuracy and Imagenet-C robustness (mCE). The robustness results for BiT and ViT are as reported by Paul & Chen (2021).

In our cross-validation of $\alpha$, we found small values less than $0.2$ perform poorly both on accuracy and mCE. Values of $\alpha$ such that $0.2 \leq \alpha \leq 0.5$ not only give the best accuracies and mCEs but also the best trade-off of mCE versus accuracy as bigger values of $\alpha$ have shown giving good values for accuracy but do not do as well on mCE. In our experiments, we found that we typically achieve good results with a frequency cutoff $c$ sampled between $[0, 1]$ as described in Algorithm 1. However, for ResNet-50 trained with a training budget that is too limited (200 instead of 600 epochs) and its smaller versions (ResNet-18 and ResNet-34), it can be beneficial to fix a minimum $c \geq \tau$ for the cutoff by sampling in the interval $[\tau, 1]$. The minimum cutoff determines the range at which band mixing will occur. We can remove band interpolation entirely and recover standard Mixup by setting $\tau = 1$. For Resnet-50 with too few training epochs, we found that a good value for the minimum is $0.1$, but we found much better results can be achieved with 600 epochs without any modications to Algorithm 1.

**Stylized-ImageNet.** We confirm that our method indeed increases both accuracy on Stylized ImageNet and the shape bias as shown in table 2. For ResNet-50, Robustmix almost doubles the shape

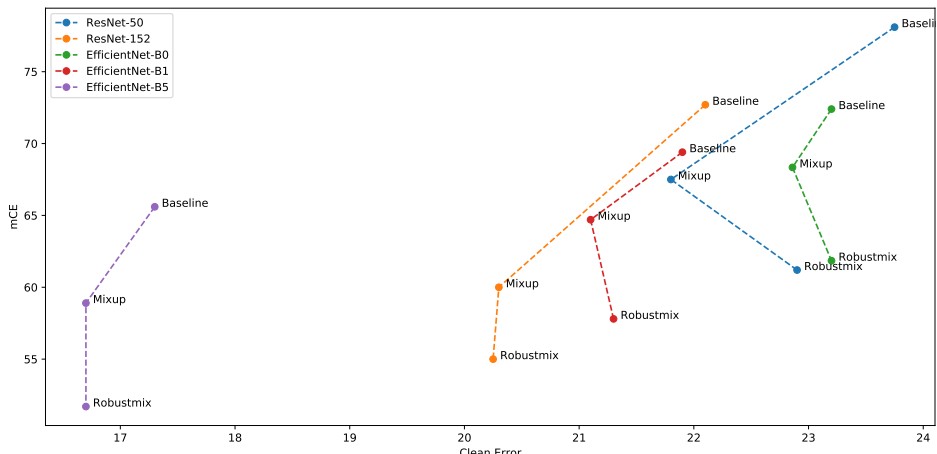

Figure 3: Highlighting the tradeoff between mCE and Clean Error for various models.

bias from baseline (from 19 to 37) and improves it by 63% over Mixup; while relative improvements on SIN accuracy are of 72% and 33% respectively over baseline and Mixup. The same observation for EfficientNet-B5 wich improves shape bias by near 50% and SIN accuracy by near 60 % over the baseline.

| Method/Parameters | SIN Accuracy | Shape Bias |
|---|---|---|
| ResNet-50 Baseline | 15.6 | 19.25 |
| ResNet-50 Mixup | 20.1 | 22.7 |
| ResNet-50 Robustmix | **26.8** | **37.0** |
| EfficientNet-B5 Baseline | 25.3 | 44.4 |
| EfficientNet-B5 Mixup | 28.75 | 48.3 |
| EfficientNet-B5 Robustmix | **40.3** | **66.1** |

Table 2: Accuracy and shape bias computed on Stylized Imagenet.

## 4.4 ADVERSARIAL PERTURBATIONS

In this section, we consider robustness to adversarial perturbations (Goodfellow et al., 2014). Adversarial perturbations make no visually distinguishable difference to the human eye but make models misclassify examples. Wang et al. (2020) and Yin et al. (2019) have shown that adversarial perturbations of unregularized models disproportionally affect higher frequencies. As we have seen in Figure 5, Robustmix encourages the model to rely more on the lower frequencies to make predictions. Our hypothesis is that this will have a beneficial effect on adversarial robustness. For our experiment we use one of the first methods proposed in the Deep Learning community to construct adversarial examples, the *"Fast Gradient Sign Method"*(FGSM) (Goodfellow et al., 2014). The adversarial example is constructed by adding a perturbation proportional to the sign of the gradient of the loss with respect to the input: $\mathbf{x} = \mathbf{x} + \epsilon \operatorname{sign}\left(\Delta_{\mathbf{x}} J(\theta, \mathbf{x}, y)\right)$. As seen in Figure 4, we find Robustmix is more robust than the baselines to this type of adversarial attack.

## 4.5 ANALYSIS AND DISCUSSION

**Low frequency bias** In order to quantify the degree to which models rely on lower frequencies, we measure how much accuracy drops as we remove higher frequency information with a low-pass filter. Figure 5 shows that Robustmix is comparatively more robust to the removal of high frequencies. This indicates that models trained with Robustmix rely significantly less on these high-frequency features to make accurate predictions.

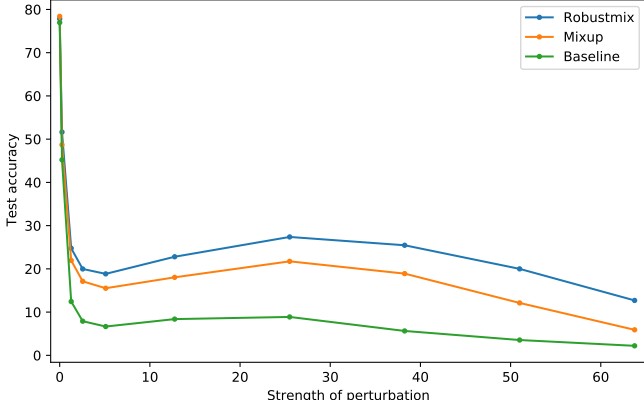

Figure 4: Test accuracy w.r.t. increasing adversarial perturbation strength. The comparison is done here on ResNet-50 models for baseline, Mixup and Robustmix.

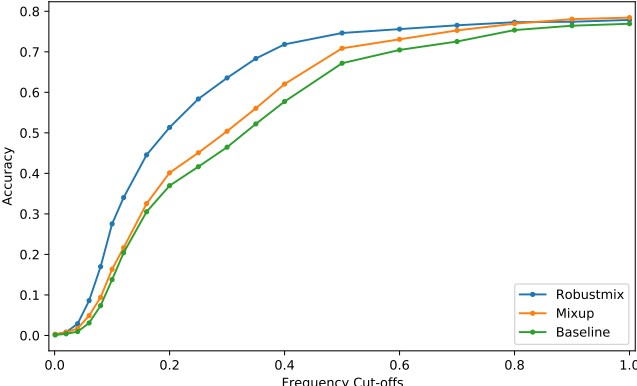

Figure 5: Test accuracy on Imagenet samples passed through a low-pass filter with increasing cut-off. As expected, we observe that Robustmix is more robust to the removal of high frequencies than Mixup. The comparison is done here on ResNet-50 models.

## 5 CONCLUSION

In this paper, we have introduced a new method to improve robustness called Robustmix which regularizes models to focus more on lower spatial frequencies to make predictions. We have shown that this method yields improved robustness on a range of benchmarks including Imagenet-C and Stylized Imagenet. In particular, this approach attains an mCE of 44.8 on Imagenet-C with Efficientnet-B8, which is competitive with models trained on $300\times$ more data.

Our method offers a promising new research direction for robustness with a number of open challenges. We have used a standard DCT based low-pass filter on images and L2 energy metric to determine the contribution of each label. This leaves many alternatives to be explored, such as: different data modalities like audio; more advanced frequency separation techniques like Wavelets; and alternative contribution metrics for mixing labels.

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
