# OpenReview forum: "Robustmix: Improving Robustness by Regularizing the Frequency Bias of Deep Nets"
_ICLR.cc/2022/Conference — ICLR 2022 Submitted_

### Official Review · Reviewer_3E2o · 2021-10-28

**Correctness:** 4
**Technical Novelty And Significance:** 2
**Empirical Novelty And Significance:** 3
**Recommendation:** 3
**Confidence:** 5

**Main Review:**

1. Strengths

    To my mind, the presented idea, while being rather simple for implementation, is well-sound with recent advances in the synthesis of signal processing and deep learning. The writing is clear and easy to understand.


2. Weaknesses.

    a. First of all, the idea of spectral mixing itself is not particularly novel. E.g., paper [1] that proposed to use it for the domain transfer problem, and [2] presented the *f-mixup* procedure for black-box adversarial attacks.

    b. Although the approach of selecting $\lambda_c$ according to the relative spectral energy looks inspiring and theoretically motivated, it needs an ablation study. How much does it outperform, e.g. mixing with the weights just proportional to $c$?

    c. The influence of two preliminary Mixup steps also was not assessed. Are they really necessary? Does it not suffice just to mix two images in the spectral domain (i.e., set $\lambda_L = 0, \, \lambda_H = 1$)?


3. Questions.

    a. Fig. 4 shows that in some cases stronger perturbation (e.g., the strength of ~5 vs ~12) leads to better test accuracy, especially in Robustmix and Mixup cases. Do the authors have any explanation for such behavior?

4. References

    [1] Yang and Soatto. FDA: Fourier Domain Adaptation for Semantic Segmentation. 2020.

    [2] Li et al. F-mixup: Attack CNNs From Fourier Perspective. 2021.

**Summary Of The Paper:**

This paper proposes a new spectral augmentation Robustmix, aiming to improve the robustness of image classifiers. In detail, the presented transformation consists of two preliminary Mixup steps and one final stage mixing low and high frequencies from the images obtained during the two initial steps. The pseudo-label is computed by weighing the labels, produced by Mixup steps, according to the relative amount of energy in low and high bands.

As the experiments show, applying Robustmix during training always leads to improved robustness to noise, corruptions, and adversarial perturbations, though in some cases trading off with a performance on vanilla datasets.

**Summary Of The Review:**

1. Pre-rebuttal score.

    Despite the interesting findings in the paper, currently, I rate the submission below the acceptance threshold. The reasons are twofold: Firstly, the idea is not completely novel. Secondly, the proper ablation study for the proposed augmentation was not conducted. I ask the authors to address this weakness during the rebuttal period.

2. Post-rebuttal update.

    After reading the authors' feedback, I keep the initial assessment. The paper misses a proper ablation study while presenting method is of limited novelty.

---

> ### Author Response · Authors · 2021-11-22
> **Thank you for your review!**
>
> Thank you for your feedback. We will use these comments to improve the paper. Please see our detailed response below.
>
> a. “First of all, the idea of spectral mixing itself is not particularly novel...”
>
> Though the frequency mixing idea alone is not particularly novel, we present a special formulation that improves on the state of the art mce measurement , a standard robustness metric, while keeping/improving on the clean accuracy. Please, note that this adaptation was not proposed before in that context.
>
> b. “Although the approach of selecting λc according to the relative spectral energy looks inspiring and theoretically motivated, it needs an ablation study. How much does it outperform, e.g. mixing with the weights just proportional to c?“
>
> To be general the idea is to weight the label of x1(c) by any function f(c) to account for the impact of cutting a cut-off c. Intuitively increasing functions f(c) are better with f(c)-->0 as c --> 0, such that cutting at a bigger c should contribute more to the final label than cutting at a smaller c.  So in this regard f(c)=c is a valid proposition that we did try as well as other functions similar to the energy ratio function in figure 2 eg. the cdf of distributions such as Pareto or Exponential which have additional parameters to be tuned. These methods didn’t perform as well, mainly in terms of robustness (mCE).
>
> c. “The influence of two preliminary Mixup steps also was not assessed. Are they really necessary? Does it not suffice just to mix two images in the spectral domain (i.e., set  λL=0,, λH=1)?“
>
> We did try simpler cases like this, some similar to what [1] does, in our early stages. The aim of our work was to improve on robustness while trying to keep at least or improve on the clean accuracy. It turned out not to perform well in that regard.
>
> d. About the behaviour mentioned about Fig. 4, we do not yet have an explanation for this. Thank you for bringing this to our attention.
>
> e. References
>    - [1] Yang and Soatto. FDA: Fourier Domain Adaptation for Semantic Segmentation. 2020.
>    - [2] Li et al. F-mixup: Attack CNNs From Fourier Perspective. 2021.
>    - [3] Fixing Data Augmentation to Improve Adversarial Robustness, 2021
>    - [4]Robust Learning Meets Generative Models: Can Proxy Distributions Improve Adversarial Robustness?, 2021
>    - [5] Smooth Adversarial Training, 2021
>    - [6] Understanding and Improving Fast Adversarial Training, 2020
>    - [7]Benchmarking neural network robustness to common corruptions and perturbations, 2018
>    - [8] Augmix: A simple data processing method to improve robustness and uncertainty, 2019
>    - [9]IMAGENET-trained CNNS are biased towards texture; increasing shape bias improves accuracy and robustness, 2019
>    - [10] https://openreview.net/pdf?id=S1gmrxHFvB on page 4 “augmentations”

---

> > ### Comment · Reviewer_3E2o · 2021-11-22
> > **Please do not hide ablations**
> >
> > First of all, thank you for your response. I would like to bring to your notice that it is strictly necessary to demonstrate in the paper that simpler modifications of your approach do not perform as well as the main model. For example, you have stated in your feedback that
> >
> > > We did try simpler cases like this, some similar to what [1] does, in our early stages... It turned out not to perform well in that regard.
> >
> > > ...we did try as well as other functions similar to the energy ratio function... These methods didn’t perform as well...
> >
> > Unless you report the results of these experiments, a reader cannot understand where the reason for the improvements lies.

---

### Official Review · Reviewer_5zdd · 2021-11-02

**Correctness:** 2
**Technical Novelty And Significance:** 3
**Empirical Novelty And Significance:** 2
**Recommendation:** 5
**Confidence:** 3

**Main Review:**

The paper is well written and the method is easy to understand. There are lots of comparison runs, and these illuminate the strengths and weaknesses of the method to some degree. However, I find the evaluation still somewhat lacking.

In particular, the method appears to not improve over standard Mixup in terms of clean accuracy, and only wins convincingly when measuring the corruption error mCE. However, these corruptions include effects such as blur and weathering that corrupt the high frequencies, so this victory is not very surprising given how Robustmix explicitly creates a low-frequency bias.

The obvious challenger to Robustmix (and Mixup, for that matter) would be augmenting training data with corruptions that resemble the ones used in the measurements, but this approach has not been measured in isolation, except for AugMix with ResNet-50. But AugMix explicitly removes the sharpness augmentation which can remove high frequencies, so the benefit of RobustMix there (in terms of mCE) could be just because of its ability to corrupt high frequencies in the training data. RandAugment provides plenty of variation to training data, but it is not tested on its own, only when combined with Robustmix. I would have hoped to see tests with RandAugment but without Robustmix to gauge their effectiveness separately. Moreover, the tests with EfficientNet-B8 include neither Mixup or RandAugment alone, or their combination. As such, these measurements are not orthogonal enough to draw firm conclusions about the relative benefits of Robustmix.

The paper also does not measure a simpler alternative where low and high frequencies are taken from different images (without mixing) and the label is decided based on spectral energy as in Robustmix. Variants like this are briefly discussed on Page 3 but dismissed without measurements. The point about encouraging linearity within a frequency band makes sense to me, but it would be nice to see that backed up with data.

The extra tests in Sections 4.4 and 4.5, related to adversarial perturbations and low-frequency bias, are very interesting. However, their usefulness is limited because they do not specify the test setup, i.e., which network was used or how it was trained. For completeness, these tests should also discuss how the results compare against state of the art. It is to be expected that methods geared explicitly for these purposes (e.g., adversarial training) are better in that regard, but it would be interesting to know how far the proposed method gets.

The paper is missing a citation to RandAugment.


**Summary Of The Paper:**

The paper proposes a novel data augmentation method for training classifiers. The idea is to mix two images in two different ways using standard Mixup, and then compose a final image by taking different frequency bands from the two mixed images. The training label is based on signal energy in different frequency bands, generally strongly favoring the label of the low-frequency content. Evaluations suggest the new scheme is more robust against image corruptions than previous methods, but not all relevant (combinations of) methods have been evaluated.

**Summary Of The Review:**

The proposed technique is well described and simple to implement. Its benefits and weaknesses are left unclear by evaluation that includes key challengers such as Mixup, AugMix, and RandAugment only in scattered combinations in different network setups. However, the obtained mCE of 44.8 in Imagenet-C is a very good result, and the paper should emphasize this by showing that no previous methods or their combinations reach that when using the same network architecture.

---

> ### Author Response · Authors · 2021-11-22
> **Thank you for your review!**
>
> Thank you for your feedback. These comments will be very helpful for the improvement of our work. Please see below our detailed response.
>
> * “... so this victory is not very surprising given how Robustmix explicitly creates a low-frequency bias”
>
> Like previous approaches (and especially on smaller models) we observe a tradeoff between robustness and accuracy. Generally, the goal is to significantly increase robustness without a significant accuracy penalty. It is indeed that most augmentations in ImageNet-C directly affect high-frequency features. We hope that eventually, an image dataset with real-world corruptions will replace the artificially generated ImageNet-C such that robustness can be more reliably measured, however introducing a completely new dataset lies outside the scope of this work. Note that Stylized-ImageNet already performs a transformation that is not a simple high-frequency transformation like blurring or random noise.
>
> * “But AugMix explicitly removes the sharpness augmentation which can remove high frequencies, so the benefit of RobustMix there (in terms of mCE) could be just because of its ability to corrupt high frequencies in the training data”.
>
> We have followed previous work [10] which avoids overlap between the augmentations in the train and test setup. The reason for this is that the goal is to measure the robustness to transformations that are unknown at training time.
>
> * “The paper also does not measure a simpler alternative….”
>
> We did try simpler cases like this, some similar to what [1] does, in our early stages. The aim of our work was to improve on robustness while trying to keep at least or improve on the clean accuracy. It turned out not to perform well in that regard.
>
>
> * Model used (ResNet-50) is now specified under the image caption. Note that all comparison graphs were done with ResNet-50 models.
>
> * “For completeness, these tests should also discuss how the results compare against state of the art…”
>
> From the overall literature, we distinguish to the best of our knowledge two schools of thought regarding robustness assessment: The first one is the accuracy measurement of models after adversarial attacks also called sometimes “robust accuracy” [2, 3, 4, 5, 6], and the second one is the mCE measurements [7, 8]. The former in general (at least in most papers encountered) do not provide mCE measurements. Of course, it would be interesting in future works to compare all these methods with a single metric. However, following the latter,  our approach puts together mCE reported in several papers and/or computed by ourselves for the sake of comparison of our method with the others from the same school of thought. In that regard, we were able to highlight both the differences and the improvements.
>
> * The missing RandAugment citation was a mistake, it is fixed.
>
> * References
>    - [1] Yang and Soatto. FDA: Fourier Domain Adaptation for Semantic Segmentation. 2020.
>    - [2] Li et al. F-mixup: Attack CNNs From Fourier Perspective. 2021.
>    - [3] Fixing Data Augmentation to Improve Adversarial Robustness, 2021
>    - [4]Robust Learning Meets Generative Models: Can Proxy Distributions Improve Adversarial Robustness?, 2021
>    - [5] Smooth Adversarial Training, 2021
>    - [6] Understanding and Improving Fast Adversarial Training, 2020
>    - [7]Benchmarking neural network robustness to common corruptions and perturbations, 2018
>    - [8] Augmix: A simple data processing method to improve robustness and uncertainty, 2019
>    - [9]IMAGENET-trained CNNS are biased towards texture; increasing shape bias improves accuracy and robustness, 2019
>    - [10] https://openreview.net/pdf?id=S1gmrxHFvB on page 4 “augmentations”

---

### Official Review · Reviewer_EYUe · 2021-11-08

**Correctness:** 3
**Technical Novelty And Significance:** 3
**Empirical Novelty And Significance:** 2
**Recommendation:** 5
**Confidence:** 4

**Main Review:**

The approach is interesting but there are still some open questions:

- Generalization of the approach: The generalization of the approach is not thoroughly discussed. The frequency sensitivity is dependent on the image type. The application of your approach e.g. to technical drawings with sharp edges will behave differently than applying it to images with smooth gradients. Furthermore, the discussion of the approach based on Stylized-ImageNet and ImageNet-C is not really convincing to discuss the quality of the approach in general. Stylized-ImageNet aims towards achieving robustness with respect to different (higher-order) textures. Hence, the basic construction metric of both approaches are related, so that similar results can be expected.

- A comparison with images “augmented” by real sensor/environment noise would be interesting for the reader. Likewise, ImageNet-C only applies very artificial noise to the image dataset, which is not really comparable to real noise, especially the very artificial snow, rain and fog effects without consideration of the depth of the image scene.
- But of course, this is a serious point of criticism of many robustness augmentation approaches being proposed the last years since most of the proposed approaches are not really free of systematic domain shifts.

- DCT calculation: it becomes not clear why tweo 1-D DCTs are separately calculated instead of a single 2-D DCT. The authors should clarify if the rationale behind this is just complexity and what is the impact w.r.t. accuracy.

Minor comments:
- Include the term mean corruption error to the paper abstract to explain the acronym mCE.



**Summary Of The Paper:**

This paper discusses a new proposal on improving the robustness of image classification/segmentation by a frequency-guided data augmentation approach. The proposed technique is based on a DCT transformation to determine frequency bands with high energy in order with respect to their sensitivity. The data augmentation will be done by a linear combination of images weighted by the energy contribution of the different energy bands.

**Summary Of The Review:**

In summary, I would encourage the authors to discuss the generalizability of the approach,before the paper can be recommended for acceptance at the conference.

---

> ### Author Response · Authors · 2021-11-22
> **Thank you for your review!**
>
> Thank you for your feedback. We will use these comments to improve the paper. Please see our detailed response below.
>
> * On the “ Generalization of the approach...”
>
> It is indeed the case that different image types like technical drawings with sharp edges could display different behavior or require re-tuning hyperparameters to work well with Robustmix. On a more general note: In practice, robustness research is constrained by the existing benchmarks and baselines available in the field. Perfect out of distribution generalization that encompasses all thinkable augmentations or image subclasses is an untestable hypothesis in practice. Our paper shows that Robustmix improves robustness for object recognition with natural images, which is an important application area.
>
> * “Furthermore, the discussion of the approach based on Stylized-ImageNet and ImageNet-C is not really convincing to discuss the quality of the approach in general…”
>
> The results presented in this paper rely on the usual benchmarking dataset and metrics for the robustness literature[7, 8, 9]. The Stylized-ImageNet benchmark is meant to distinguish between a bias towards shape or texture. We believe our results on Stylized-ImageNet complement the standard robustness results because it shows that the inductive bias is more human-like, in the sense that it is more sensitive to shape than texture.
>
> * “A comparison with images “augmented” by real sensor/environment noise would be interesting for the reader….”
>
> To the best of our knowledge, ImageNet-C is the current standard approach for comparing robustness approaches. Indeed, we agree that a dataset based on true environment noise would be strictly better than an artificial noise. However, the goal of this work is to propose a new robustness method rather than improving the metrics used to compare such methods.
>
> * “DCT calculation: it becomes not clear why two 1-D DCTs are separately calculated instead of a single 2-D DCT….”
>
>  Because the 2-D DCT is a linear and spatially separable transformation it is equivalent to performing a 1-D DCT along both spatial dimensions.
>
> * Sorry for missing to put “mean corruption error to the paper abstract to explain the acronym mCE”, it has now been added.
>
> * References
>    - [1] Yang and Soatto. FDA: Fourier Domain Adaptation for Semantic Segmentation. 2020.
>    - [2] Li et al. F-mixup: Attack CNNs From Fourier Perspective. 2021.
>    - [3] Fixing Data Augmentation to Improve Adversarial Robustness, 2021
>    - [4]Robust Learning Meets Generative Models: Can Proxy Distributions Improve Adversarial Robustness?, 2021
>    - [5] Smooth Adversarial Training, 2021
>    - [6] Understanding and Improving Fast Adversarial Training, 2020
>    - [7]Benchmarking neural network robustness to common corruptions and perturbations, 2018
>    - [8] Augmix: A simple data processing method to improve robustness and uncertainty, 2019
>    - [9]IMAGENET-trained CNNS are biased towards texture; increasing shape bias improves accuracy and robustness, 2019
>    - [10] https://openreview.net/pdf?id=S1gmrxHFvB on page 4 “augmentations”

---

### Decision · Program_Chairs · 2022-01-20

**Decision:**

Reject

**Comment:**

The paper proposes a data augmentation approach that extends Mixup with high- and low-pass filtering operations, in order to regularize deep networks towards focusing on low frequency components of the input signal.  Reviewers are unconvinced about the significance of the contribution.  Reviewer 5zdd notes that the method does not improve over standard Mixup in the absence of corruption error.  Reviewer 3E2o notes that "the idea of spectral mixing itself is not particularly novel", and also asks for ablation studies concerning the hyperparameters of the method; the author response unfortunately does not provide enough detail on ablation experiments.  The AC agrees with the reviewers and does not believe the author response has addressed weaknesses in a satisfactory manner.